# TTE-CAM: Built-in Class Activation Maps for Test-Time Explainability in Pretrained Black-Box CNNs

**Kerol Djoumessi**[1]  iD                          KEROL.DJOUMESSI-DONTEU@UNI-TUEBINGEN.DE

**Philipp Berens**[1,2]  iD                              PHILIPP.BERENS@UNI-TUEBINGEN.DE

[1] *Hertie Institute for AI in Brain Health, University of Tübingen, Germany*

[2] *Tübingen AI Center, University of Tübingen, Germany*

## Abstract

Convolutional neural networks (CNNs) achieve state-of-the-art performance in medical image analysis yet remain opaque, limiting adoption in high-stakes clinical settings. Existing approaches face a fundamental trade-off: post-hoc methods provide unfaithful approximate explanations, while inherently interpretable architectures are faithful but often sacrifice predictive performance. We introduce TTE-CAM, a test-time framework that bridges this gap by converting pretrained black-box CNNs into self-explainable models via a convolution-based replacement of their classification head, initialized from the original weights. The resulting model preserves black-box predictive performance while delivering built-in faithful explanations competitive with post-hoc methods, both qualitatively and quantitatively.

**Keywords:** Test-time explainability, Built-in CAMs, Mechanistic faithfulness, CNNs.

## 1. Introduction

Convolutional neural networks (CNNs) achieve human-level performance across many tasks, including medical image analysis (Liu et al., 2019), yet their opaque decision processes limit interpretability and hinder adoption in high-stakes clinical settings (Ratti and Graves, 2022). Existing explainability approaches face a fundamental trade-off: post-hoc methods generate saliency maps from the model that do not directly drive the output, making them inherently unfaithful (Adebayo et al., 2018). In contrast, interpretable-by-design architectures (Rudin, 2019; Djoumessi et al., 2024) are faithful—their predictions are computed from the explanation—but they often require complex training or involve a trade-off in predictive performance. Bridging this gap by transforming high-performing black-box CNNs into self-explainable models without retraining or loss of accuracy remains an open challenge.

We propose TTE-CAM, an architectural reformulation of class activation maps (CAMs) that transforms pretrained black-box CNNs into self-explainable models by replacing the classification head with $1 \times 1$ convolution layers initialized from the original weights. This reformulation yields built-in CAMs that serve as the sole input to the final prediction, enabling linearly interpretable decisions without post-hoc overhead. Unlike SoftCAM (Djoumessi and Berens, 2025), which requires retraining, and conventional CAM-based post-hoc methods that derive explanations by weighting penultimate feature maps using different methods such as classification layer weights, gradients, or perturbations (He et al., 2022), TTE-CAM integrates CAMs directly into the architecture. This preserves predictive performance while providing faithful, built-in explanations that are competitive with post-hoc approaches, as demonstrated on two medical imaging classification tasks.

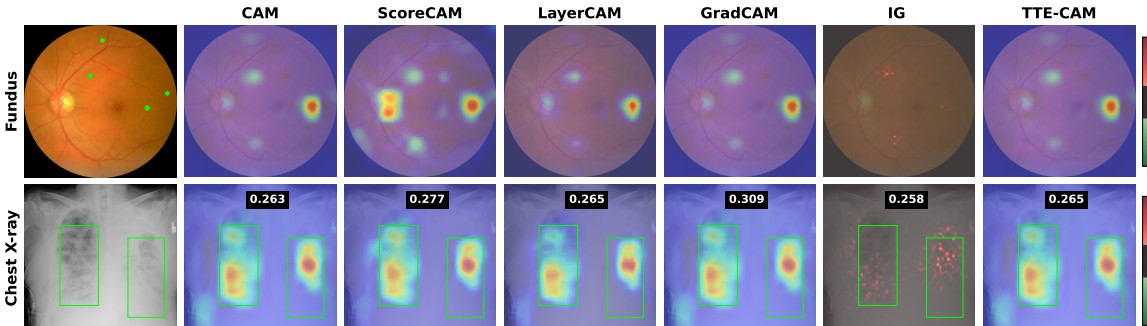

Figure 1: **Qualitative explanations comparison.** The first column shows a DR fundus with clinical annotations (green markers) and a pneumonia CXR with ground-truth bounding boxes (green). Columns 2-6 show post-hoc saliency maps; the last column shows TTE-CAM explanations. CXR scores indicate activation precision. Both samples were randomly selected from their corresponding disease classes.

## 2. Materials and Methods

**Datasets.** TTE-CAM was evaluated on two public medical imaging datasets. The Kaggle fundus Diabetic Retinopathy (DR) dataset (Dugas et al., 2015) was used for binary classification of No DR (grade 0) versus DR (grades 1–4), while the RSNA Chest X-Ray (CXR) dataset (Shih et al., 2019) was used for pneumonia detection. For explanation evaluation, RSNA bounding box annotations and clinical annotations from 65 DR fundus images (Djoumessi et al., 2025a) were used for quantitative and qualitative assessment, respectively.

**Method.** TTE-CAM reformulates the classification head of pretrained CNNs by removing the global average pooling (GAP) layer and replacing the fully connected layer (FCL) with a $1 \times 1$ convolutional layer comprising $C$ filters, where $C$ is the number of classes. Because a FCL is equivalent to a $1 \times 1$ convolution (Donteu et al., 2023), the pretrained classification weights can be transferred directly without retraining. This reformulation mirrors the original CAM architecture (Zhou et al., 2016), in which class activation maps are obtained post-hoc by weighting feature maps with classification layer weights—here integrated into the architecture. The resulting layer produces built-in CAMs that are spatially averaged to compute class scores and then passed through a softmax to obtain the final predictions.

**Post-hoc baseline.** TTE-CAM was compared against five post-hoc explainability methods from three families: gradient-free (CAM, ScoreCAM) (Zhou et al., 2016; Wang et al., 2020), gradient-based (GradCAM, LayerCAM) (Selvaraju et al., 2017; Jiang et al., 2021), and the backpropagation-based Integrated Gradients (IG) (Sundararajan et al., 2017).

**Evaluation metrics.** Predictive performance was evaluated using accuracy (Acc.) and area under the curve (AUC). Explanation quality was assessed with three metrics ($k = 10$): *top-k sensitivity* (Yeh et al., 2019), measuring the relative drop in predicted probability after masking the top-k most relevant regions; *top-k localization*, quantifying the overlap between the top-k activated regions and annotated lesions; and *activation precision* (Djoumessi and Berens, 2025), measuring the fraction of activations within ground-truth bounding boxes.

| | Metrics | CAM | S. CAM | L. CAM | G. CAM | IG | TTE-CAM |
|---|---|---|---|---|---|---|---|
| Fundus | Topk Prec. ↑ | $.33 \pm .29$ | $.28 \pm .22$ | $.31 \pm .26$ | $.39 \pm .28$ | $.39 \pm .28$ | $.33 \pm .28$ |
| | Topk Sens. ↓ | 0.629 | 0.668 | 0.644 | 0.629 | 0.605 | 0.629 |
| CXR | Acti. Prec. ↑ | $.12 \pm .09$ | $.13 \pm .10$ | $.12 \pm .09$ | $.13 \pm .10$ | $.12 \pm .09$ | $.12 \pm .09$ |
| | Topk Sens. ↓ | 0.953 | 0.959 | 0.955 | 0.953 | 0.963 | 0.953 |

Table 1: **Quantitative explanation comparison.** ↑ higher is better; ↓ lower is better.

## 3. Results

TTE-CAM was applied to a ResNet-50 (He et al., 2016) trained on each dataset, with the checkpoint achieving the best validation accuracy used at test-time[1]. Replacing the FCL with a $1 \times 1$ convolutional classifier preserved predictive performance, yielding Acc. $= 0.899$, AUC $= 0.923$ for DR and Acc. $= 0.953$, AUC $= 0.988$ for pneumonia detection.

Qualitative (Fig. 1) and quantitative (Tab. 1) results show that TTE-CAM produces explanations similar to CAM and competitive with other methods across both datasets, while being built-in by design. The sparse fundus annotations are better suited for top-k localization, whereas the denser CXR bounding boxes are better suited for activation precision.

## 4. Discussion and Conclusion

We show that pretrained back-box CNNs can provide built-in explanations at inference time by replacing the classification head with convolutional classifiers. TTE-CAM preserves the original predictive performance while producing explanations competitive with five post-hoc baselines spanning gradient-free, gradient-based, and backpropagation-based methods. Like post-hoc methods, it leverages pretrained weights without retraining, but generates explanations in a single forward pass, in contrast to post-hoc methods that require one forward pass per class (e.g. GradCAM) or multiple passes per class (e.g. ScoreCAM). Importantly, the contribution of TTE-CAM is not improved localization over CAM-based methods, but the integration of explanations directly into the prediction pipeline, eliminating dependence on gradients, perturbations, or external attribution procedures. This turns pretrained CNNs into lightweight interpretable-by-design models without retraining.

TTE-CAM explanations are identical to CAM and competitive with other baselines, sharing a related feature map weighting mechanism. Like all CAM-based methods, reliance on low-resolution feature maps can produce coarse explanations, limiting fine-grained localization, as observed in DR. Weight transfer constraints further restrict applicability to architectures where the final feature map channel dimension matches the classifier input size (e.g., ResNet and DenseNet), excluding models such as VGG. Future work could address these constraints and extend this mechanism to vision transformers (Djoumessi et al., 2025b) for built-in attention map explanations and explanation-aware clinical verification workflows.

## Acknowledgments

This project was supported by the Hertie and the German Science Foundation.

---

1. The code is available at https://github.com/kdjoumessi/Test-Time-Explainability

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
