# OpenReview forum: "TTE-CAM: Built-in Class Activation Maps for Test-Time Explainability in Pretrained Black-Box CNNs"
_MIDL.io/2026/Short_Papers — MIDL 2026 - Short Papers Poster_

### Official Review · Reviewer_5yjD · 2026-04-29
**A well-written paper proposing a single-pass explainability method**

**Rating:** 5
**Confidence:** 5

**Review:**

The authors present TTE-CAM, an explainability method for CNNs on two datasets: Kaggle fundus Diabetic Retinopathy Dataset and the RSNA Chest X-ray dataset. The proposed method is compared against five posthoc baselines spanning gradient-free, gradient-based, and backpropagation-based methods. The results are clearly presented and discussed. The main benefit of the proposed method is generating explanations in a single forward pass, in contrast to post-hoc methods that require one forward pass per class (e.g., GradCAM) or multiple passes per class (e.g., ScoreCAM).

**Summary:**

This paper introduces TTE-CAM, a test-time framework to convert pretrained black-box CNNs into self-explainable models. Their contribution is to remove the global average pooling and replace the fully connected layer with a 1x1 convolutional layer comprising *C* filters, where *C* is the number of classes.

**Strengths:**

- The work is properly positioned within the related work.
- The authors make their code publicly available on a Github repository.
- The proposed method is compared against five posthoc baselines: gradient-free, gradient-based and backpropagation-based methods.

**Weaknesses:**

- There are no clear benefits of the explainability method beyond a single forward pass, the paper does not demonstrate additional usefulness compared to existing approaches.
- The authors could clarify how the images in Figure 1 were selected.

**Justification Of Rating:**

The paper is clearly written and well presented. I recommend its acceptance as a short paper to encourage further discussion within the community.

---

### Decision · Program_Chairs · 2026-05-08

Accept (Poster)